# General Table Completion using a Bayesian Nonparametric Model

**Isabel Valera**
Department of Signal Processing
and Communications
University Carlos III in Madrid
`ivalera@tsc.uc3m.es`

**Zoubin Ghahramani**
Department of Engineering
University of Cambridge
`zoubin@eng.cam.ac.uk`

## Abstract

Even though heterogeneous databases can be found in a broad variety of applications, there exists a lack of tools for estimating missing data in such databases. In this paper, we provide an efficient and robust table completion tool, based on a Bayesian nonparametric latent feature model. In particular, we propose a general observation model for the Indian buffet process (IBP) adapted to mixed continuous (real-valued and positive real-valued) and discrete (categorical, ordinal and count) observations. Then, we propose an inference algorithm that scales linearly with the number of observations. Finally, our experiments over five real databases show that the proposed approach provides more robust and accurate estimates than the standard IBP and the Bayesian probabilistic matrix factorization with Gaussian observations.

## 1 Introduction

A full 90% of all the data in the world has been generated over the last two years and this expansion rate will not diminish in the years to come [17]. This extreme availability of data explains the great investment that both the industry and the research community are expending in data science. Data is usually organized and stored in databases, which are often large, noisy, and contain missing values. Missing data may occur in diverse applications due to different reasons. For example, a sensor in a remote sensor network may be damaged and transmit corrupted data or even cease to transmit; participants in a clinical study may drop out during the course of the study; or users of a recommendation system rate only a small fraction of the available books, movies, or songs. The presence of missing values can be challenging when the data is used for reporting, information sharing and decision support, and as a consequence, missing data treatment has captured the attention in diverse areas of data science such as machine learning, data mining, and data warehousing and management.

Several studies have shown that probabilistic modeling can help to estimate missing values, detect errors in databases, or provide probabilistic responses to queries [19]. In this paper, we exclusively focus on the use of probabilistic modeling for missing data estimation, and assume that the data are missing completely at random (MCAR). There is extensive literature in probabilistic missing data estimation and imputation in homogeneous databases, where all the attributes that describe each object in the database present the same (continuous or discrete) nature. Most of the work assumes that databases contain only either continuous data, usually modeled as Gaussian variables [21], or discrete, that can be either modeled by discrete likelihoods [9] or simply treated as Gaussian variables [15, 21]. However, there still exists a lack of work dealing with heterogeneous databases, which in fact are common in real applications and where the standard approach is to treat all the attributes, either continuous or discrete, as Gaussian variables. As a motivating example, consider a database that contains the answers to a survey, including diverse information about the participants such as age (count data), gender (categorical data), salary (continuous non negative data), etc.

In this paper, we provide a general Bayesian approach for estimating and replacing the missing data in heterogeneous databases (being the data MCAR), where the attributes describing each object can be either discrete, continuous or mixed variables. Specifically, we account for real-valued, positive real-valued, categorical, ordinal and count data. To this end, we assume that the information in the database can be stored in a matrix (or table), where each row corresponds to an object and the columns are the attributes that describe the different objects. We propose a novel Bayesian nonparametric approach for general table completion based on feature modeling, in which each object is represented by a set of latent variables and the observations are generated from a distribution determined by those latent features. Since the number of latent variables needed to explain the data depends on the specific database, we use the Indian buffet process (IBP) [8], which places a prior distribution over binary matrices where the number of columns (latent variables) is unbounded. The standard IBP assumes real-valued observations combined with conjugate likelihood models that allow for fast inference algorithms [4]. Here, we aim at dealing with heterogeneous databases, which may contain mixed continuous and discrete observations.

We propose a general observation model for the IBP that accounts for mixed continuous and discrete data, while keeping the properties of conjugate models. This allows us to propose an inference algorithm that scales linearly with the number of observations. The proposed algorithm does not only infer the latent variables for each object in the table, but it also provides accurate estimates for its missing values. Our experiments over five real databases show that our approach for table completion outperforms, in terms of accuracy, the Bayesian probabilistic matrix factorization (BPMF) [15] and the standard IBP which assume Gaussian observations. We also observe that the approach based on treating mixed continuous and discrete data as Gaussian fails in estimating some attributes, while the proposed approach provides robust estimates for all the missing values regardless of their discrete or continuous nature.

The main contributions in this paper are: i) A general observation model (for mixed continuous and discrete data) for the IBP that allows us to derive an inference algorithm that scales linearly with the number of objects, and its application to build ii) a general and scalable tool to estimate missing values in heterogeneous databases. An efficient C-code implementation for Matlab of the proposed table completion tool is also released on the authors website.

## 2 Related Work

In recent years, probabilistic modeling has become an attractive option for building database management systems since it allows estimating missing values, detecting errors, visualizing the data, and providing probabilistic answers to queries [19]. BayesDB,[1] for instance, is a database management system that resorts to Crosscat [18], which originally appeared as a Bayesian approach to model human categorization of objects. BayesDB provides missing data estimates and probabilistic answer to queries, but it only considers Gaussian and multinomial likelihood functions.

In the literature, probabilistic low-rank matrix factorization approaches have been broadly applied to table completion (see, e.g., [14, 15, 21]). In these approaches, the table database $\mathbf{X}$ is approximated by a low-rank matrix representation $\mathbf{X} \approx \mathbf{ZB}$, where $\mathbf{Z}$ and $\mathbf{B}$ are usually assumed to be Gaussian distributed. Most of the works in this area have focused on building automatic recommendation systems, which appears as the most popular application of missing data estimation [14, 15, 21]. More specific models to build recommendation systems can be found in [7, 22], where the authors assume that the rates each user assign to items are generated by a probabilistic generative model which, based on the available data, accounts for similarities among users and among items to provide good estimates of the missing rates.

Probabilistic matrix factorization can also be viewed as latent feature modeling, where each object is represented by a vector of continuous latent variables. In contrast, the IBP and other latent feature models (see, e.g., [16]) assume binary latent features to represent each object. Latent feature models usually assume homogeneous databases with either real [14, 15, 21] or categorical data [9, 12, 13], and only a few works consider heterogeneous data, such as mixed real and categorical data [16]. However, up to our knowledge, there are no general latent feature models (nor table completion tools) to directly deal with heterogeneous databases. To fill this gap, in this paper we provide a general table completion approach for heterogeneous databases, based on a generalized IBP, that allows for efficient inference.

# 3 Model Description

Let us assume a table with $N$ objects, where each object is defined by $D$ attributes. We can store the data in an $N \times D$ observation matrix $\mathbf{X}$, in which each $D$-dimensional row vector is denoted by $\mathbf{x}_n = [x_n^1, \ldots, x_n^D]$ and each entry is denoted by $x_n^d$. We consider that column vectors $\mathbf{x}^d$ (i.e., each dimension in the observation matrix $\mathbf{X}$) may contain the following types of data:

- Continuous variables:
  1. Real-valued, i.e., $x_n^d \in \Re$
  2. Positive real-valued, i.e., $x_n^d \in \Re_+$.
- Discrete variables:
  1. Categorical data, i.e., $x_n^d$ takes values in a finite unordered set, e.g., $x_n^d \in \{$'blue', 'red', 'black'$\}$.
  2. Ordinal data, i.e., $x_n^d$ takes values in a finite ordered set, e.g., $x_n^d \in \{$'never', 'sometimes', 'often', 'usually', 'always'$\}$.
  3. Count data, i.e., $x_n^d \in \{0, \ldots, \infty\}$,

We assume that each observation $x_n^d$ can be explained by a $K$-length vector of latent variables associated to the $n$-th data point $\mathbf{z}_n = [z_{n1}, \ldots, z_{nK}]$ and a weighting vector[2] $\mathbf{B}^d = [b_1^d, \ldots, b_K^d]$ (being $K$ the number of latent variables), whose elements $b_k^d$ weight the contribution of $k$-th the latent feature to the $d$-th dimension of $\mathbf{X}$. We gather the latent binary feature vectors $\mathbf{z}_n$ in a $N \times K$ matrix $\mathbf{Z}$, which follows an IBP with concentration parameter $\alpha$, i.e., $\mathbf{Z} \sim \mathrm{IBP}(\alpha)$ [8]. We place a Gaussian distribution with zero mean and covariance matrix $\sigma_B^2 \mathbf{I}_K$ over the weighting vectors $\mathbf{B}^d$. For convenience, $\mathbf{z}_n$ is a K-length row vector, while $\mathbf{B}^d$ is a K-length column vector.

To accommodate for all kinds of observed random variables described above, we introduce an auxiliary Gaussian variable $y_n^d$, such that when conditioned on the auxiliary variables, the latent variable model behaves as a standard IBP with Gaussian observations. In particular, we assume $y_n^d$ is Gaussian distributed with mean $\mathbf{z}_n \mathbf{B}^d$ and variance $\sigma_y^2$, i.e.,

$$p(y_n^d | \mathbf{z}_n, \mathbf{B}^d) = \mathcal{N}(y_n^d | \mathbf{z}_n \mathbf{B}^d, \sigma_y^2),$$

and assume that there exists a transformation function over the variables $y_n^d$ to obtain the observations $x_n^d$, mapping the real line $\Re$ into the observation space. The resulting generative model is shown in Figure 1, where $\mathbf{Z}$ is the IBP latent matrix, and $\mathbf{Y}^d$ and $\mathbf{B}^d$ contain, respectively, the auxiliary Gaussian variables $y_n^d$ and the weighting factors $b_k^d$ for the $d$-dimension of the data. Additionally, $\Psi^d$ denotes the set of auxiliary random variables needed to obtain the observation vector $\mathbf{x}^d$ given $\mathbf{Y}^d$, and $\mathcal{H}^d$ contains the hyper-parameters associated to the random variables in $\Psi^d$. This model assumes that the observations $x_n^d$ are independent given the latent matrix $\mathbf{Z}$, the weighting matrices $\mathbf{B}^d$ and the auxiliary variables $\Psi^d$. Therefore, the likelihood can be factorized as

$$p(\mathbf{X} | \mathbf{Z}, \{\mathbf{B}^d, \Psi^d\}_{d=1}^D) = \prod_{d=1}^D p(\mathbf{x}^d | \mathbf{Z}, \mathbf{B}^d, \Psi^d) = \prod_{d=1}^D \prod_{n=1}^N p(x_n^d | \mathbf{z}_n, \mathbf{B}^d, \Psi^d).$$

Note that, if we assume Gaussian observations and set $\mathbf{Y}^d = \mathbf{x}^d$, this model resembles the standard IBP with Gaussian observations [8]. In addition, conditioned on the variables $\mathbf{Y}^d$, we can infer the latent matrix $\mathbf{Z}$ as in the standard IBP. We also remark that auxiliary Gaussian variables to link a latent model with the observations have been previously used in Gaussian processes for multi-class classification [6] and for ordinal regression [2]. However, up to our knowledge, this simple approach has not been used to account for mixed continuous and discrete data, and the existent approaches for the IBP with discrete observations propose non-conjugate likelihood models and approximate inference algorithms [12, 13].

## 3.1 Likelihood Functions

Now, we define the set of transformations that map from the Gaussian variables $y_n^d$ to the corresponding observations $x_n^d$. We consider that each dimension in the table $\mathbf{X}$ may contain any of the discrete or continuous variables detailed above, provide a likelihood function for each kind of data and, in turn, also a likelihood function for mixed data.

**Real-valued Data.** In this case, we assume that $\mathbf{x}^d = \mathbf{Y}^d$ in the model in Figure 1 and consider the standard approach when dealing with real-valued observations, which consist of assuming a Gaussian likelihood function. In particular, as in the standard linear-Gaussian IBP [8], we assume that each observation $x_n^d$ is distributed as

$$p(x_n^d|\mathbf{z}_n, \mathbf{B}^d) = \mathcal{N}(x_n^d|\mathbf{z}_n\mathbf{B}^d, \sigma_y^2).$$

**Positive Real-valued Data.** In order to obtain positive real-valued observations, i.e., $x_n^d \in \Re_+$, we apply a transformation over $y_n^d$ that maps from the real numbers to the positive real numbers, i.e.,

$$x_n^d = f(y_n^d + u_n^d),$$

where $u_n^d$ is a Gaussian noise variable with variance $\sigma_u^2$, and $f : \Re \to \Re_+$ is a monotonic differentiable function. By change of variables, we obtain the likelihood function for positive real-valued observations as

$$p(x_n^d|\mathbf{z}_n, \mathbf{B}^d) = \frac{1}{\sqrt{2\pi(\sigma_y^2 + \sigma_u^2)}} \exp\left\{-\frac{1}{2(\sigma_y^2 + \sigma_u^2)}(f^{-1}(x_n^d) - \mathbf{z}_n\mathbf{B}^d)^2\right\} \left|\frac{d}{dx_n^d}f^{-1}(x_n^d)\right|, \quad (1)$$

where $f^{-1} : \Re_+ \to \Re$ is the inverse function of the transformation $f(\cdot)$, i.e, $f^{-1}(f(v)) = v$. Note that in this case we resort to the Gaussian variable $u_n^d$ in order to obtain $x_n^d$ from $y_n^d$, and therefore, $\Psi^d = u_d^d$ and $\mathcal{H}^d = \sigma_u^2$.

**Categorical Data.** Now we account for categorical observations, i.e., each observation $x_n^d$ can take values in the unordered index set $\{1, \ldots, R_d\}$. Hence, assuming a multinomial probit model, we can write

$$x_n^d = \arg\max_{r \in \{1, \ldots, R_d\}} y_{nr}^d, \quad (2)$$

being $y_{nr}^d \sim \mathcal{N}(y_{nr}^d|\mathbf{z}_n\mathbf{b}_r^d, \sigma_y^2)$ where $\mathbf{b}_r^d$ denotes the K-length weighting vector, in which each $b_{kr}^d$ weights the influence of the $k$-th feature for the observation $x_n^d$ taking value $r$. Note that, under this likelihood model, since we have a Gaussian auxiliary variable $y_{nr}^d$ and a weighting factor $b_{kr}^d$ for each possible value of the observation $r \in \{1, \ldots, R_d\}$, we need to gather all the weighting factors $b_{kr}^d$ in a $K \times R_d$ matrix $\mathbf{B}^d$, and all the Gaussian auxiliary variables $y_{nr}^d$ in the $N \times R_d$ matrix $\mathbf{Y}^d$.

Under this observation model, we can write $y_{nr}^d = \mathbf{z}_n\mathbf{b}_r^d + u_{nr}^d$, where $u_{nr}^d$ is a Gaussian noise variable with variance $\sigma_y^2$, and therefore, we can obtain the probability of each element $x_n^d$ taking value $r \in \{1, \ldots, R_d\}$ as [6]

$$p(x_n^d = r|\mathbf{z}_n, \mathbf{B}^d) = \mathbb{E}_{p(u)}\left[\prod_{\substack{j=1 \\ j \neq r}}^{R_d} \Phi\left(u + \mathbf{z}_n(\mathbf{b}_r^d - \mathbf{b}_j^d)\right)\right], \quad (3)$$

where subscript $r$ in $\mathbf{b}_r^d$ states for the column in $\mathbf{B}^d$ ($r \in \{1, \ldots, R_d\}$), $\Phi(\cdot)$ denotes the cumulative density function of the standard normal distribution and $\mathbb{E}_{p(u)}[\cdot]$ denotes expectation with respect to the distribution $p(u) = \mathcal{N}(0, \sigma_y^2)$.

**Ordinal Data.** Consider ordinal data, in which each element $x_n^d$ takes values in the ordered index set $\{1, \ldots, R_d\}$. Then, assuming an ordered probit model, we can write

$$x_n^d = \begin{cases} 1 & \text{if } y_n^d \leq \theta_1^d \\ 2 & \text{if } \theta_1^d < y_n^d \leq \theta_2^d \\ \quad \vdots \\ R_d & \text{if } \theta_{R_d-1}^d < y_n^d \end{cases} \quad (4)$$

where again $y_n^d$ is Gaussian distributed with mean $\mathbf{z}_n\mathbf{B}^d$ and variance $\sigma_y^2$, and $\theta_r^d$ for $r \in \{1, \ldots, R_d - 1\}$ are the thresholds that divide the real line into $R_d$ regions. We assume the thresholds $\theta_r^d$ are sequentially generated from the truncated Gaussian distribution $\theta_r^d \propto \mathcal{N}(\theta_r^d|0, \sigma_\theta^2)\mathbb{I}(\theta_r^d > \theta_{r-1}^d)$, where $\theta_0^d = -\infty$ and $\theta_{R_d}^d = +\infty$. As opposed to the categorical case, now we have a unique

weighting vector $\mathbf{B}^d$ and a unique Gaussian variable $y_n^d$ for each observation $x_n^d$. Hence, the value of $x_n^d$ is determined by the region in which $y_n^d$ falls.

Under the ordered probit model [2], the probability of each element $x_n^d$ taking value $r \in \{1, \ldots, R_d\}$ can be written as

$$p(x_n^d = r | \mathbf{z}_n, \mathbf{B}^d) = \Phi\left(\frac{\theta_r^d - \mathbf{z}_n \mathbf{B}^d}{\sigma_y}\right) - \Phi\left(\frac{\theta_{r-1}^d - \mathbf{z}_n \mathbf{B}^d}{\sigma_y}\right). \tag{5}$$

Let us remark that, if the $d$-dimension of the observation matrix contains ordinal data, the set of auxiliary variables reduces to the Gaussian thresholds $\Psi^d = \{\theta_1^d, \ldots, \theta_{R_d-1}^d\}$ and $\mathcal{H}^d = \sigma_\theta^2$.

**Count Data.** In count data each observation $x_n^d$ takes non-negative integer values, i.e., $x_n^d \in \{0, \ldots, \infty\}$. Then, we assume

$$x_n^d = \lfloor f(y_n^d) \rfloor, \tag{6}$$

where $\lfloor v \rfloor$ returns the floor of $v$, that is the largest integer that does not exceed $v$, and $f : \Re \rightarrow \Re_+$ is a monotonic differentiable function that maps from the real numbers to the positive real numbers. We can therefore write the likelihood function as

$$p(x_n^d | \mathbf{z}_n, \mathbf{B}^d) = \Phi\left(\frac{f^{-1}(x_n^d + 1) - \mathbf{z}_n \mathbf{B}^d}{\sigma_y}\right) - \Phi\left(\frac{f^{-1}(x_n^d) - \mathbf{z}_n \mathbf{B}^d}{\sigma_y}\right) \tag{7}$$

where $f^{-1} : \Re_+ \rightarrow \Re$ is the inverse function of the transformation $f(\cdot)$.

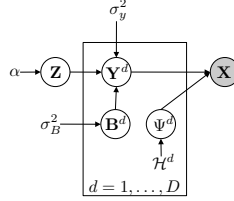

Figure 1: Generalized IBP for mixed continuous and discrete observations.

## 4   Inference Algorithm

In this section we describe our algorithm for inferring the latent variables given the observation matrix. Under our model, detailed in Section 3, the probability distribution over the observation matrix is fully characterized by the latent matrices $\mathbf{Z}$ and $\{\mathbf{B}^d\}_{d=1}^D$ (as well as the auxiliary variables $\Psi^d$). Hence, if we assume the latent vector $\mathbf{z}_n$ for the $n$-th datapoint and the weighting factors $\mathbf{B}^d$ (and the auxiliary variables $\Psi^d$) to be known, we have a probability distribution over missing observations $x_n^d$ from which we can obtain estimates for $x_n^d$ by sampling from this distribution,[3] or by simply taking either its mean, mode or median value. However, this procedure requires the latent matrix $\mathbf{Z}$ and the latent weighting factors $\mathbf{B}^d$ (and $\Psi^d$) to be known.

We use Markov Chain Monte Carlo (MCMC) methods, which have been broadly applied to infer the IBP matrix (see, e.g., in [8, 23, 20]). The proposed inference algorithm is summarized in Algorithm 1. This algorithm exploits the information in the available data to learn the similarities among the objects (captured in our model by the latent feature matrix $\mathbf{Z}$), and how these latent features show up in the attributes that describe the objects (captured in our model by $\mathbf{B}^d$). In Algorithm 1, we first need to update the latent matrix $\mathbf{Z}$. Note that conditioned on $\{\mathbf{Y}^d\}_{d=1}^D$, both the latent matrix $\mathbf{Z}$ and the weighting matrices $\{\mathbf{B}^d\}_{d=1}^D$ are independent of the observation matrix $\mathbf{X}$. Additionally, since $\{\mathbf{B}^d\}_{d=1}^D$ and $\{\mathbf{Y}^d\}_{d=1}^D$ are Gaussian distributed, we can analytically marginalize out the weighting matrices $\{\mathbf{B}^d\}_{d=1}^D$ to obtain $p(\{\mathbf{Y}^d\}_{d=1}^D | \mathbf{Z})$. Therefore, to infer the matrix $\mathbf{Z}$, we can apply the collapsed Gibbs sampler which presents better mixing properties than the uncollapsed

**Algorithm 1** Inference Algorithm.

---

**Input: X**
**Initialize:** initialize $\mathbf{Z}$ and $\{\mathbf{Y}^d\}_{d=1}^D$
 1: **for** each iteration **do**
 2:    Update $\mathbf{Z}$ given $\{\mathbf{Y}^d\}_{d=1}^D$.
 3:    **for** $d = 1, \ldots, D$ **do**
 4:       Sample $\mathbf{B}^d$ given $\mathbf{Z}$ and $\mathbf{Y}^d$ according to (8).
 5:       Sample $\mathbf{Y}^d$ given $\mathbf{X}$, $\mathbf{Z}$ and $\mathbf{B}^d$ (as shown in the Supplementary Material).
 6:       Sample $\Psi^d$ if needed (as shown in the Supplementary Material).
 7:    **end for**
 8: **end for**
**Output**: $\mathbf{Z}$, $\{\mathbf{B}^d\}_{d=1}^D$ and $\{\Psi^d\}_{d=1}^D$

---

Gibbs sampler and, in consequence, is the standard method of choice in the context of the standard linear-Gaussian IBP [8]. However, this algorithm suffers from a high computational cost (being complexity per iteration cubic with the number of data points $N$), which is prohibitive when dealing with large databases. In order to solve this limitation, we resort to the accelerated Gibbs sampler [4] instead. This algorithm presents linear complexity with the number of datapoints and is detailed in the Supplementary Material.

Second, we need to sample the weighting factors in $\mathbf{B}^d$, which is a $K \times R_d$ matrix in the case of categorical attributes, and a K-length column vector otherwise. We denote each column vector in $\mathbf{B}^d$ by $\mathbf{b}_r^d$. The posterior over the weighting vectors are given by

$$p(\mathbf{b}_r^d | \mathbf{y}_r^d, \mathbf{Z}) = \mathcal{N}(\mathbf{b}_r^d | \mathbf{P}^{-1}\boldsymbol{\lambda}_r^d, \mathbf{P}^{-1}), \tag{8}$$

where $\mathbf{P} = \mathbf{Z}^\top\mathbf{Z} + 1/\sigma_B^2\mathbf{I}_k$ and $\boldsymbol{\lambda}_r^d = \mathbf{Z}^\top\mathbf{y}_r^d$. Note that the covariance matrix $\mathbf{P}^{-1}$ depend neither on the dimension $d$ nor on $r$, so we only need to invert the $K \times K$ matrix $\mathbf{P}$ once at each iteration. We describe in the Supplementary Material how to efficiently compute $\mathbf{P}$ after changes in the $\mathbf{Z}$ matrix by rank one updates, without the need of computing the matrix product $\mathbf{Z}^\top\mathbf{Z}$.

Once we have updated $\mathbf{Z}$ and $\mathbf{B}^d$, we sample each element in $\mathbf{Y}^d$ from the distribution $\mathcal{N}(y_{nr}^d | \mathbf{z}_n\mathbf{b}^d, \sigma_y^2)$ if the observation $x_n^d$ is missing, and from the posterior $p(y_{nr}^d | x_n^d, \mathbf{z}_n, \mathbf{b}^d)$ specified in the Supplementary Material, otherwise. Finally, we sample the auxiliary variables in $\Psi^d$ from their posterior distribution (detailed in the Supplementary Material) if necessary. This two latter steps involve, in the worst case, sampling from a doubly truncated univariate normal distribution (see the Supplementary Material for further details), for which we make use of the algorithm in [11].

## 5 Experimental evaluation

We now validate the proposed algorithm for table completion on five real databases, which are summarized in Table 1. The datasets contain different numbers of instances and attributes, which cover all the discrete and continuous variables described in Section 3. We compare, in terms of predictive log-likelihood, the following methods for table completion:

- The proposed general table completion approach denoted by GIBP (detailed in Section 3).
- The standard linear-Gaussian IBP [8] denoted by SIBP, treating all the attributes as Gaussian.
- The Bayesian probabilistic matrix factorization approach [15] denoted by BPMF, that also treats all the attributes in $\mathbf{X}$ as Gaussian distributed.

For the GIBP, we consider for the real positive and the count data the following transformation, that maps from the real numbers to the real positive numbers, $f(x) = \log(\exp(wx) + 1)$, where $w$ is a user hyper-parameter. Before running the SIBP and the BPMF methods we normalize each column in matrix $\mathbf{X}$ to have zero-mean and unit-variance. Then, in order to provide estimates for the missing data, we denormalize the inferred Gaussian variable. Additionally, since both the SIBP and the BPMF assume continuous observations, when dealing with discrete data, we estimate each missing value as the closest integer value to the (denormalized) Gaussian variable.

| Dataset | N | D | Description |
|---|---|---|---|
| Statlog German credit dataset [5] | 1,000 | 20 (10 C + 4 O + 6 N) | Collects information about the credit risks of the applicants. |
| QSAR biodegradation dataset [10] | 1,055 | 41 (2 R + 17 P + 4 C + 18 N) | Contains molecular descriptors of biodegradable and non-biodegradable chemicals. |
| Internet usage survey dataset [1] | 1,006 | 32 (23 C + 8 O + 1 N) | Contains the responses of the participants to a survey related to the usage of internet. |
| Wine quality Dataset [3] | 6,497 | 12 (11 P + 1 N) | Contains the results of physicochemical tests realized to different wines. |
| NESARC dataset [13] | 43,000 | 55 C | Contains the responses of the participants to a survey related to personality disorders. |

Table 1: Description of datasets. 'R' states for real-valued variables, 'P' for positive real-valued variables, 'C' for categorical variables, 'O' for ordinal variables and 'N' for count variables

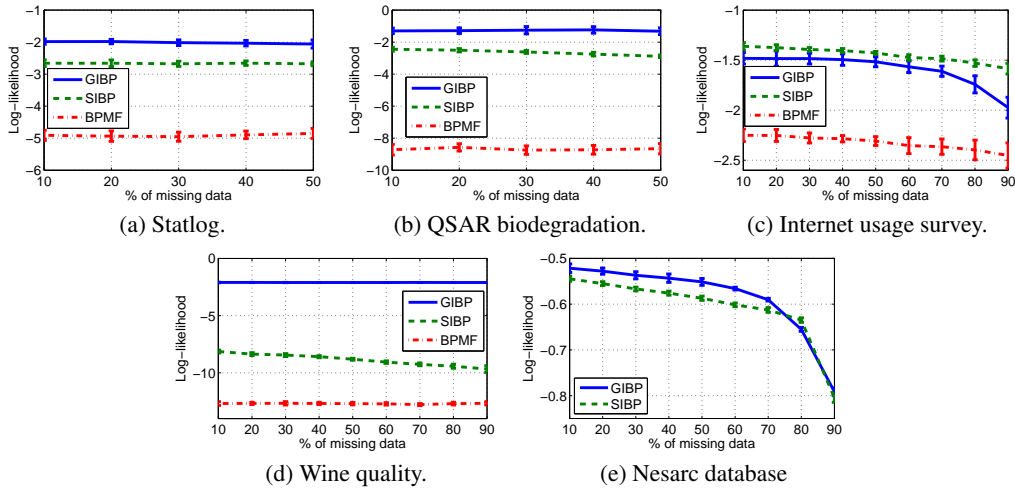

(a) Statlog.  (b) QSAR biodegradation.  (c) Internet usage survey.

(d) Wine quality.  (e) Nesarc database

Figure 2: Average test log-likelihood per missing datum. The 'whiskers' show a standard deviations from the average test log-likelihood.

In Figure 2, we plot the average predictive log-likelihood per missing value as a function of the percentage of missing data. Each value in Figure 2 has been obtained by averaging the results in 20 independent sets where the missing values have been randomly chosen. In Figures 2a and 2b, we cut the plot in 50% because, in these two databases, the discrete attributes present a mode value that is present for more than 80% of the instances. As a consequence, the SIBP and the BPMF algorithms assign probability close to one to the mode, which results in an artificial increase in the average test log-likelihood for larger percentages of missing data. For the BPMF model, we have used different numbers of latent features (in particular, 10, 20 and 50), although we only show the best results for each database, specifically, $K = 10$ for the NESARC and the wine databases, and $K = 50$ for the remainder. Both the GIBP and the SIBP have not inferred a number of (binary) latent features above 25 in any case. Note that in Figure 2e, we only plot the test log-likelihood for the GIBP and the SIBP because the BPMF provides much lower values. As expected, we observe in Figure 2 that the average test log-likelihood decreases for the three models when the number of missing values increases (flat shape of the curves are due to the y-axis scale). In this figure, we also observe that the proposed general IBP model outperforms the SIBP and the BPMF for four of the the databases, being the SIBP slightly better for the Internet database. The BPMF model presents the lowest test-log-likelihood in all the databases.

Now, we analyze the performance of the three models for each kind of discrete and continuous variables. Figure 3 shows average predictive likelihood per missing value for each attribute in the table, i.e., for each dimension in $\mathbf{X}$. In this figure we have grouped the dimensions according to the kind of data that they contain, showing in the x-axis the number of considered categories for the case of categorical and ordinal data. In this figure, we observe that the GIBP presents similar performance

for all the attributes in the five databases, while for the SIBP and the BPMF models, the test-log-likelihood falls drastically for some of the attributes, being this effect worse in the case of the BPMF (it explains the low log-likelihood in Figure 2). This effect is even more evident in Figures 2b and 2d. We also observe, in Figures 2 and 3, that both IBP based approaches (the GIBP and the SIBP) outperform the BPMF, with the proposed GIBP being the one that best performs across all the databases. We can conclude that, unlike to the BPMF and the GIBP, the GIBP provides accurate estimates for the missing data regardless of their discrete or continuous nature.

## 6 Conclusions

In this paper, we have proposed a table completion approach for heterogeneous databases, based on an IBP with a generalized likelihood that allows for mixed discrete and continuous data. We have then derived an inference algorithm that scales linearly with the number of observations. Finally, our experimental results over five real databases have shown than the proposed approach outperforms, in terms of robustness and accuracy, approaches that treat all the attributes as Gaussian variables.

(a) Statlog.

(b) QSAR biodegradation.

(c) Internet usage survey.

(d) Wine quality.

(e) Nesarc database

Figure 3: Average test log-likelihood per missing datum in each dimension of the data with 50% of missing data. In the x-axis 'R' states for real-valued variables, 'P' for positive real-valued variables, 'C' for categorical variables, 'O' for ordinal variables and 'N' for count variables. The number that accompanies the 'C' or 'O' corresponds to the number of categories.

**Acknowledgments**

Isabel Valera acknowledge the support of *Plan Regional-Programas I+D* of *Comunidad de Madrid* (AGES-CM S2010/BMD-2422), *Ministerio de Ciencia e Innovación* of Spain (project DEIPRO TEC2009-14504-C02-00 and program Consolider-Ingenio 2010 CSD2008-00010 COMONSENS). Zoubin Ghahramani is supported by the EPSRC grant EP/I036575/1 and a Google Focused Research Award.

## Footnotes

[1] http://probcomp.csail.mit.edu/bayesdb/

[2]For convenience, we capitalized here the notation for the weighting vectors $\mathbf{B}^d$.

[3]Note that sampling from this distribution might be computationally expensive. In this case, we can easily obtain samples of $x_n^d$ by exploiting the structure of our model. In particular, we can simply sample the auxiliary Gaussian variables $y_n^d$ given $z_n$ and $\mathbf{B}^d$, and then obtain an estimate for $x_n^d$ by applying the corresponding transformation, detailed in Section 3.1.

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
