[Supplementary Material]

# Supplementary Material: General Table Completion using a Bayesian Nonparametric Model

**Isabel Valera**
Department of Signal Processing
and Communications
University Carlos III in Madrid
ivalera@tsc.uc3m.es

**Zoubin Ghahramani**
Department of Engineering
University of Cambridge
zoubin@eng.cam.ac.uk

## 1 Accelerated Gibbs Sampler

In [1], the authors presented a linear-time accelerated Gibbs sampler for conjugate IBP models that effectively marginalized over the latent factors. The per-iteration complexity of this algorithm is $\mathcal{O}(N(K^2 + KD))$, which is comparable to the uncollapsed linear-Gaussian IBP sampler that has per-iteration complexity $\mathcal{O}(NDK^2)$ but does not marginalize over the weighting factors, and as a result, presents slower convergence rate.

This algorithm exploits the Bayes rule to avoid the cubic complexity with $N$ due to the computation of the marginal likelihood in the Collapsed Gibbs sampler. In particular, it applies the Bayes rule to obtain the probability of each element in the latent feature matrix $\mathbf{Z}$ being active as

$$p(z_{nk} = 1|\{\mathbf{Y}^d\}_{d=1}^D, \mathbf{Z}_{\neg nk}) \propto \frac{m_{\neg n,k}}{N} \prod_{d=1}^D \prod_{r=1}^{S_d} \int_{\mathbf{b}_r^d} p(y_{nr}^d|\mathbf{z}_n, \mathbf{b}_r^d) p(\mathbf{b}_r^d|\mathbf{y}_{\neg nr}^d \mathbf{Z}_{\neg n}) d\mathbf{b}_r^d, \quad (1)$$

where $S_d$ is the number of columns in matrices $\mathbf{Y}^d$ and $\mathbf{B}^d$ (being $S_d$ the number of categories $R_d$ for those dimension $d$ that contains categorical attributes, and $S_d = 1$ otherwise), $\mathbf{Z}_{\neg n}$ corresponds to matrix $\mathbf{Z}$ after removing the the $n$-th row, the vector $\mathbf{y}_{\neg nr}^d$ is the $r-$th column of matrix $\mathbf{Y}^d$ without the element $y_{nr}^d$, and $p(\mathbf{b}_r^d|\mathbf{x}_{\neg n}^d, \mathbf{Z}_{\neg n})$ is the posterior of $\mathbf{b}_r^d$ computed without taking the $n$-th datapoint into account, i.e.,

$$p(\mathbf{b}_r^d|\mathbf{y}_{\neg nr}^d, \mathbf{Z}_{\neg n}) = \mathcal{N}(\mathbf{b}_r^d|\mathbf{P}_{\neg n}^{-1}\boldsymbol{\lambda}_{\neg nr}^d, \mathbf{P}_{\neg n}^{-1}), \quad (2)$$

where $\mathbf{P}_{\neg n} = \mathbf{Z}_{\neg n}^\top \mathbf{Z}_{\neg n} + 1/\sigma_B^2 \mathbf{I}_K$ and $\boldsymbol{\lambda}_{\neg ny}^d = \mathbf{Z}_{\neg n}^\top \mathbf{y}_{\neg nr}^d$ are the natural parameters of the Gaussian distribution.

Note that, opposite to the notation in [1], we here resort to the natural parameters for the Gaussian distribution over the posterior of $\mathbf{b}_r^d$ instead of the mean and the covariance matrix. This formulation allows us to compute the full posterior over the weighting factors as

$$p(\mathbf{b}_r^d|\mathbf{y}_r^d, \mathbf{Z}) = \mathcal{N}(\mathbf{b}_r^d|\mathbf{P}^{-1}\boldsymbol{\lambda}_r^d, \mathbf{P}^{-1}), \quad (3)$$

where $\mathbf{P} = \mathbf{P}_{\neg n} + \mathbf{z}_n^\top \mathbf{z}_n$ and $\boldsymbol{\lambda}_r^d = \boldsymbol{\lambda}_{\neg nr}^d + \mathbf{z}_n^\top y_{nr}^d$ are the natural parameters of the Gaussian distribution.

The Accelerated Gibbs sampling algorithm iteratively samples the value of each element $z_{nk}$ according to

$$p(z_{nk} = 1|\{\mathbf{Y}^d\}_{d=1}^D, \mathbf{Z}_{\neg nk}) \propto \frac{m_{\neg n,k}}{N} \prod_{d=1}^D \prod_{r=1}^{S_d} \mathcal{N}(y_{nr}^d|\mathbf{z}_n \boldsymbol{\lambda}_{\neg nr}^d, \mathbf{z}_n \mathbf{P}_{\neg n} \mathbf{z}_n^\top + \sigma_y^2). \quad (4)$$

After having sampled all elements $z_{nk}$ for the $K_+$ non-zero columns in $\mathbf{Z}$ for each data point $n$, the algorithm samples from a distribution (where the prior is a Poisson distribution with mean $\alpha/N$) a number of new features necessary to explain that data point.

## 2    Posterior distribution over $\mathbf{Y}^d$

As previously described, in the 5-th step of Algorithm **??**, we need to sample from the auxiliary Gaussian variables $y_{nr}^d$ from the posterior distribution $p(y_{nr}^d|x_n^d, \mathbf{z}_n, \mathbf{b}^d)$. The posterior distribution $y_{nr}^d$ for all the considered types of data are given given by:

1. For real-valued observation:

$$p(y_{n1}^d|x_n^d, \mathbf{z}_n, \mathbf{B}^d) = \delta(x_n^d) \tag{5}$$

2. For positive real-valued observations:

$$p(y_{n1}^d|x_n^d, \mathbf{z}_n, \mathbf{B}^d)$$
$$= \mathcal{N}\left(y_{n1}^d \left| \left(\frac{(\mathbf{z}_n\mathbf{b}_1^d)}{\sigma_y^2} + \frac{f^{-1}(x_n^d)}{\sigma_u^2}\right)\left(\frac{1}{\sigma_y^2} + \frac{1}{\sigma_u^2}\right)^{-1}, \left(\frac{1}{\sigma_y^2} + \frac{1}{\sigma_u^2}\right)^{-1}\right). \tag{6}$$

3. For categorical observations:

$$p(y_{nr}^d|x_n^d = T, \mathbf{z}_n, \mathbf{B}^d)$$
$$= \begin{cases} \mathcal{N}(y_{nr}^d|\mathbf{z}_n\mathbf{b}_r^d, \sigma_y^2)\mathbb{I}(y_{nr}^d > \max_{j\neq r}(y_{nj}^d)) & \text{If} \quad r = T \\ \mathcal{N}(y_{nr}^d|\mathbf{z}_n\mathbf{b}_r^d, \sigma_y^2)\mathbb{I}(y_{nr}^d < y_{nT}^d) & \text{If} \quad r \neq T \end{cases} \tag{7}$$

   In words, if $x_n^d = T = r$ we sample $y_{nr}^d$ from a Gaussian truncated by the left by $\max_{j\neq r}(y_{nj}^d)$ and, otherwise, we sample from a Gaussian truncated by the right by $y_{nr}^d$ with $r = x_n^d$. Note that sampling from the variables $y_{nr}^d$ corresponds to solve a multinomial probit regression problem. To achieve identifiability we assume, without loss of generality, that the regression function $f_{R_d}(\mathbf{z}_n)$ is identically zero, and therefore, we fix $b_{kR_d}^d = 0$ for all $k$.

4. For ordinal observations:

$$p(y_{n1}^d|x_n^d = r, \mathbf{z}_n, \mathbf{B}^d) \sim \mathcal{N}(y_{n1}^d|\mathbf{z}_n\mathbf{b}_1^d, \sigma_y^2)\mathbb{I}(\theta_{r-1}^d < y_{n1}^d \leq \theta_r^d). \tag{8}$$

   Note that in this case, we also need to sample the values for the thresholds $\theta_r^d$ with $r = 1, \dots, R_d - 1$ as

$$\begin{aligned} p(\theta_r^d|y_{n1}^d) \sim &\mathcal{N}(\theta_r^d|0, \sigma_\theta^2)\mathbb{I}(\theta_r^d > \max(\theta_{r-1}^d, \max_n(y_{n1}^d|x_n^d = r)) \\ &\times \mathbb{I}(\theta_r^d < \min(\theta_r^d, \min_n(y_{n1}^d|x_n^d = r + 1)). \end{aligned} \tag{9}$$

   In this case, sampling from the variables $y_{n1}^d$ corresponds to solve an ordered probit regression problem, where the thresholds $\{\theta_r\}_{r=1}^{R_d}$ are unknown. Hence, to achieve identifiability we need to set the one of the thresholds, $\theta_1$ in our case, to a fixed value.

5. For count observations:

$$p(y_{n1}^d|x_n^d, \mathbf{z}_n, \mathbf{B}^d) = \mathcal{N}(y_{n1}^d|\mathbf{z}_n\mathbf{b}_1^d, \sigma_y^2)\mathbb{I}(f^{-1}(x_n^d) \leq y_{n1}^d < f^{-1}(x_n^d + 1)), \tag{10}$$

   where $f^{-1} : \Re_+ \rightarrow \Re$ is the inverse function of $f$, i.e., $f^{-1}(f(y)) = y$. Therefore, $y_{n1}^d$ from a Gaussian truncated by the left by $f^{-1}(x_n^d)$ and by the right by $f^{-1}(x_n^d + 1)$.

## References

[1] F. Doshi-Velez and Z. Ghahramani. Accelerated sampling for the indian buffet process. In *Proceedings of the 26th Annual International Conference on Machine Learning*, ICML '09, pages 273–280, New York, NY, USA, 2009. ACM.