[Reviews · NeurIPS 2014]

Submitted by Assigned_Reviewer_1

The paper describes a Bayesian hierarchical model model for handling mixed-type missing data (i.e., datasets that involve both continuous and discrete data) in large databases. The model relies on the use of latent Gaussian variables whose correlation is modeled using a bilinear latent factor model. Uncertainty on the number of latent factors is accounted for using an Indian Buffet process prior on the factor indicators.

General comments:

1) Although the paper does not discuss the issue explicitly, their model treats the missingness mechanism (which determines the probability that a given value is missing) as ignorable. This is unlikely to be the case in most of the databases considered in the illustration, which is a well known to be a serious issue (a classic reference is Rubin 1976, but there is an extensive statistics literature on the topic over the last 40 years). The whole article misses the point that even a very flexible model will lead very poor imputations if the probability that a value is missing depends of the characteristics of the object. Since the authors do not even attempt to justify their assumption of ignorability, I have to assume that they are unaware of the issue. I consider this a critical flaw in their work.

2) I would also like to note that the use of latent Gaussian variables to unify the treatment of mixed-type multivariate observations has quite a long history in Bayesian methods. Furthermore, the authors do not discuss how to choose the transformation to be used for positive real or count data (f in the author’s notation). In their evaluation the make ad-hoc choices, but there is absolutely no justification for it or guidelines on how to make the choice in more general cases.

3) The “nonparametric” nature of the model presented by the authors refers to its ability to automatically estimate the number of latent factors. However, the model is strongly parametric in the sense that it involves strong assumptions about the distributions of the outcomes (e.g., Gaussianity/log-Gaussianity for continuous outcomes, probit link functions for categorical data). The parametric assumption on the joint distribution of the (latent) variables is bound to have a more dramatic effect on the imputations that a slight misspecification in the number of latent factors. Again, the incremental improvement advocated by the authors seems to be focused on an aspect of the model that seems to be less critical for the application.

4) I have concerns related to the use of log-likelihoods for comparing models (for example, they do not penalize for complexity, which would of course favor the nonparametric models).

Clarity: The paper is well written and clear.

Quality: Unfortunately I am quite negative about the quality of the manuscript. I think the authors focus too much on creating a fancy model and too little on the underlying issues associated with their application.

Originality: The methods seem to be novel in the context of the computer science literature on the topic (I cannot tell for sure since I am not a computer scientist), but from the point of view of the model or the more general missing-data literature the paper is derivative.

Significance: Missing data is certainly an important problem in statistics and machine learning. However, I do not believe that the paper would have an impact.

Rubin D.B, (1976) “Inference and missing data” Biometrika: 63, 581-592.

Summary: Missing data is certainly an important problem in statistics and machine learning. However, the methods described are not really novel and they focus on issues that are somewhat tangential to those that are key for missing-data imputation.

Submitted by Assigned_Reviewer_9

In the manuscript "General Table Completion using a Bayesian Nonparametric Model" the authors propose a method for database completion using the Indian buffet process (IBP) which is a Bayesian nonparametric prior on binary matrices with a fixed number of rows and an unbounded number of columns (which correspond to features of the data associated with the rows). The authors suppose that the columns of the database fall into the following 5 classes: reals, nonnegative reals, discrete, ordinal or integer. For each of these classes, the authors provide a latent IBP model linking the entries of the database with latent features for each row. They find the latent features and likelihood parameters using standard MCMC methods for the IBP and apply their model to predicting missing entries in 5 diverse datasets ranging in size from 20k to 2m entries. This model is important for the NIPS community because it constitutes a very general application of Bayesian nonparametrics. Recently in the machine learning community there has been lots of interest in automatic statistics (see Lloyd et al. "Automatic construction and natural-language description of nonparametric regression models" 2014).

In each of the 5 classes, the latent features interact with the likelihood through a Gaussian random variable which is passed through a link function. The covariance matrix of the Gaussian random variable is learned, as are the parameters of the link functions. As the authors point out, this approach is not incredibly innovative (the IBP has been used to model each of these classes separately before, as can be seen by the citations in the manuscript). The novel aspect of the authors' work is a uniform framework in which all of these classes are combined in order to do inference on an arbitrary database. Further, in the experiments I think that the authors give quite good results defending the idea that adding the link functions (i.e., treating the classes differently instead of just versions of Gaussian random variables that just happen to be observed at positive or at integral values) really does add more predictive accuracy. Another innovative aspect of the manuscript was in the Gibbs updates, which build upon the IBP sampler from Chu and Ghahramani "Gaussian processes for ordinal regression" (2005) which marginalizes over the latent feature assignments of the data items. The authors extend this method to also marginalize over the weighting factors of the features, improving the efficiency of the algorithm.

Overall, I thought that this manuscript was good and that it should be accepted. The datasets they applied their model to are new and their results do show that learning link functions for IBP based methods improves upon standard Gaussian models. I did, however think that they should have compared with more methods. For example, they could have compared with methods based on Dirichlet process mixtures such as CrossCat (the likelihoods and link functions used in this paper could also be used to augment the CrossCat model (Shafto et al. 2011) in a way such that it becomes appropriate for the same sort of mixed-class matrices).

This manuscript still left some questions about inference based on the IBP model open: the IBP has symmetries arising from simultaneous transformations of the latent feature weights and also the feature assignments. For example, if the sign of all the entries in the feature weight matrix change, and if the feature matrix is simultaneously inverted (so 0s go to 1s and 1s go to 0s) then the likelihood of the data remains unchanged. Because modes in MCMC are attractive, this leads to slower mixing when the MCMC state is `the same distance' between two equivalent modes, it is not pulled strongly to either of them. Unfortunately, the inference method used by the authors did not address this.

In the introduction of the manuscript, the authors mention that Bayesian methods can be used to detect errors in databases (i.e., entries where there has been some corruption or mistake in data entry). But I found that they did not actually explore this in the paper. I do think that this is an excellent direction for Bayesian nonparametrics so I was disappointed that it wasn't picked up upon again in the manuscript after the introduction. They could also look at detection of the class of a column, which might not always be obvious by looking at the entries of the column alone. For example, in a database it might turn out that the discrete classes of a column actually have an ordinal relation. Also, a bunch of -9.0000's in a column of real numbers might suggest that those entries are actually missing. Or, a column with both marked missing entries and entries with 0.0000, might actually have the 0.0000 entries through a default, missing value also.

I felt like the experiments conducted by the authors demonstracted clearly that using link functions over improves the accuracy. However, I did wish that the authors explored the inference on the experiments more. For example, why does the SIBP (which is an IBP with purely Gaussian assumptions) method outperform the author's method for the `internet usage' dataset? Also, by examining the columns in Figure 3(c) for example, it is clear that there is some structure in the order in which the columns are registered in the database. Why does the BPMF method perform poorly on the first and last columns of the Nesarc dataset?
Summary: A database completion using the Indian buffet process. The novel aspect of the authors' work is a uniform framework in which many classes of data (integer, discrete, real, nonnegative) are combined in order to do inference on an arbitrary database.

Submitted by Assigned_Reviewer_16

This paper introduces a generalized linear-Gaussian likelihood for the IBP to handle matrix completion for matrices whose entries may be continuous/categorical/ordinal/etc. Non continuous data is handled by transformations of a Gaussian distributed latent variable Y. Inference for the IBP draw Z is performed with accelerated Gibbs for the IBP and Gibbs conditionals for Y take the forms of truncated univariate Gaussians. The model is compared to two continuous matrix factorization models (linear-Gaussian IBP) and Bayesian probabilistic matrix factorization, giving improvements for most datasets on predictive log-likelihood, and giving improvements on all datasets when restricting attention to non-continuous entries.

The paper is clear and well organized. The modelling contribution is some what iterative, but is to my knowledge novel. The simple and efficient inference algorithm makes the paper more attractive. Matrix completion is an important problem to the NIPS community and so I believe this paper should be accepted.

One minor issue: the (bolded) variables B^d and b^d appear to be used interchangeably, for example lies 133 and 139 use b^d as if it were B^d
Summary: The paper offers a simple extension of the linear-Gaussian IBP model to handle non-continuous data. An efficient Gibbs inference scheme is provided and the experiments show improvements over models that assuming continuous matrix entries.
Author Feedback
Author rebuttal: We thank the Reviewers for their comments and suggestions, which we believe will be very useful to improve our paper and our line of research. We belief that this paper provides a simple but useful approach for probabilistic estimation of missing data in heterogeneous databases that, as shown in our experiments, provided more robust results than the standard approach in which all the data are treated as Gaussian variables.

Specific answer for Reviewer 1
We agree with this reviewer that the issue of the missing data mechanism is important, and should have been mentioned in the introduction to the paper. We are well aware of work by Rubin (e.g. 1987) and others on this topic, and have worked on this topic ourselves, and we will clarify that the model under consideration assumes that the missing data mechanism is MCAR. This is an easy fix to the text and references, not in our opinion a critical flaw. Extending this work to model the missing data mechanism as well, to handle not missing at random (NMAR) or censored data would be very interesting future work.

We also agree the reviewer that further discussion about the selection of function f would improve the paper, and therefore, we will include it in the final version of the paper. The general framework holds for any monotonic differentiable functions, but obviously we had to make some choices to be able to run experiments. Nevertheless, we would like to point out that the easy use of the proposed model in which, as shown in our experiments, a general choice of a monotonic differentiable function f, provides robust results in five real databases, even though one could make an add-hoc choice for each database to improve the results, or perhaps infer the form of f from data.

As pointed out by the reviewer, the proposed model is nonparametric only in the number of latent features, and we do not claim the opposite in the text. The main restriction in the likelihood models that we propose for each type of the considered data lie in the fact that we need to keep the properties of conjugate models to be able to derive an efficient inference algorithm that handles tens of thousands of observations. Fully nonparametric outcome distributions would (a) not be as scalable, and (b) can often be seen as equivalent to a latent­variable + strongly parametric model (for example a DP mixture of Gaussians can be viewed as a parametric Gaussian conditioned on a single countably­infinite categorical latent variable).

Regarding our evaluation using likelihoods, crucially, note that we are always reporting predictive log­likelihood (i.e. log probability on test or held­out data, conditioned on observed data). Since this is a calibrated and honest Bayesian (or prequential) measure of test performance, unlike the training set log­likelihood, it is not necessary or in fact principled to penalize for model complexity. We agree that had we reported training log likelihoods, we would have needed to use a complexity penalty such as AIC or BIC (although these are hard to compute for nonparametric models, with effectively infinitely many parameters!).

Specific answer for Reviewer 16
Thank the reviewer for your comments, we will carefully revise the paper in order to solve any mistake or typo, as the one pointed out in your review (i.e. the interchangeably of variables B^d and b^d), the paper may contain.

Specific answer for Reviewer 9
Thank the reviewer for comments and suggestions, which we believe will be very useful to improve our paper and our line of research. We like the idea of augmenting the CrossCat model to account for mixed discrete and continuous data, and we will take it into account for future work. Since, up to our knowledge, there are no general latent feature models (nor table completion tools) to directly deal with heterogeneous databases, we decide to compare with the BPMF and the Standard IBP model both treating the data as Gaussian variables, because we believe this is the standard practical approach when dealing with missing data estimation in heterogeneous databases. Additionally, we agree the reviewer in the fact that detecting errors in databases (i.e., entries where there has been some corruption or mistake in data entry) appears as an interesting and challenging problem and, as a consequence, we will study how our proposed model perform in this task in future works.

Finally, we would like to clarify the reviewers doubts about our experimental results. First of all, we want to mention that, as shown in Figure 3, the BPMF provided comparable results to the SIBP for most of the attributes of the data in the five datasets. However, for other attributes, it assign probabilities close to zeros to some of the test observations, which makes the average test log-­likelihood to fall drastically. We also observe that the proposed GIBP outperforms the SIBP for most of the databases but for the internet usage, where the GIBP has not been able to properly capture some of the attributes (mainly the ordinal ones) in the database. It might be due to either the pre­process of the data or the hyperparameter selection. Perhaps, a larger parameter alpha would help to increase the number of latent features, making the model more accurate. Hence, we also consider as a future minor extension to include a new step in the MCMC sampler, to infer the IBP parameter alpha, or perhaps to generalise to the two­- and three­-parameter IBP (e.g. Teh & Gorur, 2009).